# FedDCT: Towards Interference-Free One-Shot Federated Learning via Frequency Domain Aggregation

## Abstract

The widespread adoption of Large Language Models (LLMs) has introduced a significant challenge for federated learning (FL): their massive parameter counts make the traditional multi-round FL approach prohibitively expensive due to high communication costs. One-shot federated learning (OFL) has emerged as a promising solution by limiting communication to a single round. However, Unlike standard FL, which can mitigate the effects of non-IID data over multiple communication rounds, OFL must simultaneously address "amplified spatial heterogeneity" and "parameter space collisions" in a single update. The lack of iterative communication means there's no opportunity to progressively resolve these conflicts, leading to parameter interference that can severely degrade the global model's performance. To address this, we propose the one-shot Federated Frequency Separated aggregation (`FedDCT`) method. This novel framework for LLMs uses the discrete cosine transform (DCT) to construct orthogonal parameter spaces. This allows clients to operate independently with minimal collisions, facilitating effective model adaptation without the need for iterative communication, even with heterogeneous data. Through extensive experiments, we demonstrate that `FedDCT` outperforms existing one-shot methods while maintaining comparable communication efficiency to non-federated approaches.[1]

## 1 Introduction

Large Language Models (LLMs) have revolutionized natural language processing (NLP), demonstrating remarkable capabilities across a diverse range of tasks (Brown et al., 2020; Chowdhery et al., 2022; Touvron et al., 2023a). However, the current centralized training paradigm raises significant privacy concerns, especially when dealing with sensitive data such as personal conversations, medical records and proprietary business information (Bommasani et al., 2021; Carlini et al., 2021). Federated Learning (FL) offers a promising alternative by enabling collaborative model training without exposing private data (Kairouz et al., 2021; Yang et al., 2020; Ren et al., 2025; Fan et al., 2025). In typical FL settings, clients train models locally and periodically exchange model updates with a server, which aggregates these updates to improve a global model.

Despite its privacy benefits, traditional FL faces a critical challenge when applied to training LLMs: the immense communication overhead. With modern LLMs containing hundreds of billions of parameters (Brown et al., 2020; Chowdhery et al., 2022), each communication round requires transferring enormous amounts of data—potentially hundreds of gigabytes per client. This becomes especially problematic in bandwidth-constrained environments, where network capacity might be limited to just 1-10 MB/s (Yang et al., 2020; Ren et al., 2025; Tang et al., 2023). For instance, transmitting the full parameter set of a 175B-parameter model (approximately 700GB) would require nearly 20 hours at 10 MB/s for each round, making multi-round FL training impractical.

One-shot federated learning (OFL) has emerged as a potential solution by limiting communication to a single round (Guha et al., 2019; Zhang et al., 2022; Li et al., 2021). In this paradigm, clients train models locally and communicate with the server only once, drastically reducing the communication burden.

---

[1]Code is available at on https://anonymous.4open.science/r/FEDFFT-555F.

However, The fundamental challenge of one-shot federated learning lies in its inability to leverage iterative refinement to resolve client heterogeneity conflicts, forcing all reconciliation to occur within a single aggregation step (Zhou et al., 2020; Zhang et al., 2022). Unlike standard FL, which can gradually mitigate the effects of non-IID data distributions across multiple communication rounds, one-shot FL must simultaneously address amplified spatial heterogeneity and parameter space collisions in a single update. This constraint is particularly problematic when client data distributions vary significantly, as the local updates from different clients not only reflect conflicting data characteristics but also compete for overlapping regions in the parameter space during aggregation. The absence of iterative communication rounds eliminates the opportunity to detect and resolve these conflicts progressively, resulting in parameter interference that can severely distort the global model's ability to generalize across diverse client distributions and downstream tasks, thus raising an important question:

*How to effective resolve the severe parameter interference in **single** round in Federated Learning?*

In this paper, we introduce `FedDCT`, a novel one-shot federated learning framework for LLMs that effectively addresses the challenges of heterogeneity and parameter interference. `FedDCT` leverages the natural orthogonality properties of the 2D Discrete Cosine Transform (DCT) with sparse frequency sampling to enable flexible control over client update orthogonality. By supporting both sampling without replacement (strict orthogonality) and sampling with replacement (statistical orthogonality), our approach uniquely enables truly independent client operation—each client can select frequency components without any knowledge of others' choices or coordination with the server. This maintain minimal interference through the DCT's quadratic expansion of the parameter space from $d \times d$ to $d^2$ frequencies, reducing the expected collision proportion from $\mathcal{O}(|\Omega|^2/d)$ in traditional methods to $\mathcal{O}(|\Omega|^2/d^2)$. Through theoretical analysis and extensive experimentation, we demonstrate the effectiveness of our approach for OFL of LLMs across diverse language understanding tasks. Our main contributions are as follows:

- We propose a novel frequency-based aggregation method that exploits the 2D DCT's quadratic parameter space expansion to achieve minimal collision rates while preserving federated learning's privacy principles. Critically, our sampling with replacement strategy allows clients to operate independently without coordination, accommodating heterogeneous devices and data distributions.

- We formalize the parameter space collision problem in one-shot federated learning and provide a theoretical framework distinguishing between strict orthogonality (requiring coordination) and statistical orthogonality (enabling independence), showing DCT-based aggregation achieves both.

- We empirically validate our approach on benchmark NLP tasks, demonstrating significant improvements over existing one-shot FL methods, while requiring only a fraction of the communication cost compared to traditional federated learning in multi-round settings.

## 2 RELATED WORK

Approaches for mitigating FL communication overhead can be broadly divided into two main categories: 1) Parameter-Efficient Fine-Tuning (PEFT) for language models, which reduces trainable parameters, and 2) OFL, which limits communication to a single round.

**PEFT:** The massive size of LLMs has driven substantial research in PEFT methods, which adapt pre-trained models by updating only a small subset of parameters. Low-Rank Adaptation (LoRA) (Hu et al., 2021) decomposes weight updates into low-rank matrices, reducing the number of trainable parameters from $\mathcal{O}(d^2)$ to $\mathcal{O}(rd)$, where $r \ll d$ is the chosen rank. Adapter-based approaches (Houlsby et al., 2019; Pfeiffer et al., 2020; Chen et al., 2022) insert trainable modules within transformer architectures while keeping pre-trained weights frozen. Prefix tuning (Li & Liang, 2021) and prompt tuning (Lester et al., 2021) prepend trainable vectors to the input or intermediate layers. These approaches have demonstrated effectiveness, often matching full fine-tuning performance while updating less than 1% of the parameters.

While these methods reduce the computational and memory requirements for adaptation, they were primarily designed for centralized learning settings. When applied to FL, they still face significant challenges related to communication efficiency and data heterogeneity. Recent work (Zhang et al.,

2023a;b) has begun exploring PEFT methods in multi-round FL. However, their application in the one-shot setting remains largely unexplored, particularly for LLMs.

**OFL:** OFL methods limit communication to a single round to minimize bandwidth requirements. Guha et al. (Guha et al., 2019) introduced this concept with two approaches: 1) selectively ensembling client models, and 2) distilling client knowledge using a public dataset. Follow-up works have expanded on these ideas, but continue to face limitations. Data-dependent approaches (Guha et al., 2019; Li et al., 2021) require auxiliary public datasets, which might be unavailable or inadequate for specialized domains. Synthetic data generation methods (Zhou et al., 2020; Zhang et al., 2022) use GANs or other generative models, but struggle with complex data distributions. Parameter aggregation approaches (Wang et al., 2020; Yurochkin et al., 2019) directly combine client model weights, but typically perform poorly under heterogeneous data distributions. Zhou et al. (Zhou et al., 2020) explored dataset distillation for one-shot FL, but found performance degrading under moderate to high data heterogeneity. Recently, Jhunjhunwala et al. (Jhunjhunwala et al., 2023) provided theoretical analysis for one-shot averaging with over-parameterized networks, but did not address the challenges of heterogeneous data. The fundamental limitation of existing OFL approaches is their inability to combine client knowledge without interference, especially as the number of clients increases. This challenge becomes more acute with LLMs, where the parameter space is vast but still finite relative to the number of potential clients and the complexity of language data distributions.

# 3 PRELIMINARIES

## 3.1 ONE-SHOT FEDERATED LEARNING FOR PEFT

**Parameter Efficient Federated Learning for LLMs.** Given $\kappa$ FL clients $\{\mathcal{C}_\kappa\}_{\kappa=1}^K$ and a global FL server $\mathcal{S}$, client $\mathcal{C}_\kappa$ train on the local dataset $(\mathcal{X}_\kappa, \mathcal{Y}_\kappa)$ via optimizing $\operatorname{argmin}_{\Delta \boldsymbol{W}_\kappa} \mathcal{L}(\boldsymbol{W} + \Delta \boldsymbol{W}_\kappa; \mathcal{X}_\kappa, \mathcal{Y}_\kappa)$ where $\boldsymbol{W}$ is the parameters of the base model and $\Delta \boldsymbol{W}_\kappa$ is the local update. The server aggregates all uploaded parameter updates through weighted averaging:

$$\Delta \bar{\boldsymbol{W}} = \sum_{\kappa=1}^K \gamma_\kappa \Delta \boldsymbol{W}_\kappa, \text{ where } \gamma_\kappa = \frac{|\mathcal{X}_\kappa|}{\sum_{\kappa=1}^K |\mathcal{X}_\kappa|}, \tag{1}$$

and update the global model in each communication round $t$ as:

$$\boldsymbol{W}^t = \boldsymbol{W}^{t-1} + \Delta \bar{\boldsymbol{W}}. \tag{2}$$

**One-Shot Federated Learning for PEFT.** Traditional FL requires multiple communication rounds ($t \gg 1$) to achieve good performance, which becomes prohibitively expensive for large models. One-shot FL addresses this by limiting the process to a single communication round. In this setting, clients train locally on their datasets and communicate with the server exactly once. Instead, the global model is updated just once ($t = 1$):

$$\boldsymbol{W}^1 = \boldsymbol{W}^0 + \Delta \bar{\boldsymbol{W}} \tag{3}$$

## 3.2 THE AGGREGATION METHODS

**Parameter Efficient Aggregation for One-Shot Federated Learning.** To reduce communication cost, recent works (Hu et al., 2021) decompose parameter updates $\Delta \boldsymbol{W}$ via low-rank adaptation[1]:

$$\Delta \bar{\boldsymbol{W}} = \sum_{\kappa=1}^K \Delta \boldsymbol{W}_\kappa, \ \ \Delta \boldsymbol{W}_\kappa = \boldsymbol{A}_\kappa \boldsymbol{B}_\kappa, \tag{4}$$

where $\boldsymbol{A}_\kappa \in \mathbb{R}^{d \times r}$, $\boldsymbol{B}_\kappa \in \mathbb{R}^{r \times k}$, and $r \ll \min\{d, k\}$. This reduces communication cost from $\mathcal{O}(dk)$ to $\mathcal{O}(r(d + k))$. Existing aggregation strategies for LoRA, however, face fundamental limitations:

- **FedAvg**: FedIT (Zhang et al., 2024) proposes averaging local LoRA modules via FedAvg to reduce resource requirements during training $\Delta \bar{\boldsymbol{W}}_{\text{avg}} = \bar{\boldsymbol{A}}\bar{\boldsymbol{B}}$, where $\bar{\boldsymbol{A}} = \sum_{\kappa=1}^K \boldsymbol{A}_\kappa$, $\bar{\boldsymbol{B}} = \sum_{\kappa=1}^K \boldsymbol{B}_\kappa$. However, this introduces undesirable cross-terms $\boldsymbol{A}_i \boldsymbol{B}_j$ ($i \neq j$), leading to parameter interference that distorts client-specific knowledge—especially under heterogeneous data distributions.

---

[1]We omit the $\gamma_\kappa^t$ for simplicity.

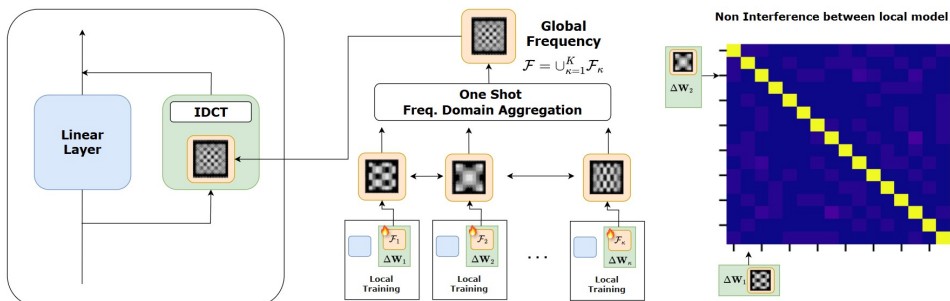

Figure 1: `FedDCT` partitions client updates into orthogonal subspaces via 2D Discrete Cosine Transform (DCT). The diagram illustrates how individual client updates are transformed into the frequency domain, where they operate on distinct subsets of frequency components. These frequency-separated updates are then aggregated by the server and reconstructed back into the parameter space using the Inverse Discrete Cosine Transform (IDCT) to form the final global model update. The right-hand side of the figure visually demonstrates the non-interference between local models in this frequency-based approach.

.

- **FedStack**: FLoRA (Wang et al., 2024b) proposes stacking separately averaged LoRA modules to construct the global model, thereby preserving client-specific updates. $\bar{A} = [A_1, \ldots, A_K]$, $\bar{B} = [B_1, \ldots, B_K]$. While this prevents cross-terms mathematically, it still causes functional interference when columns of different $A_\kappa$ matrices learn similar features, creating redundancy and competition during inference as they operate on overlapping regions of the parameter space.

To address the fundamental shortcomings of existing aggregation methods, we introduce a novel parameter efficient aggregation strategy specifically designed for one-shot federated learning.

## 4 THE PROPOSED `FedDCT` METHOD

Our key insight is to eliminate parameter interference by operating in the frequency domain, where we can naturally assign orthogonal frequency components to different clients. Specifically, we design to decompose the global model update as:

$$\Delta \boldsymbol{W} = \sum_{\kappa=1}^{K} \Delta \boldsymbol{W}_\kappa = \sum_{\kappa=1}^{K} \mathtt{iDCT}(\mathcal{F}_\kappa) \tag{5}$$

where $\mathtt{iDCT}(\cdot)$ denote inverse transformation of 2D DCT and each client $\kappa$ operates exclusively on a distinct subset of frequency components $\mathcal{F}_\kappa = \{F(u,v)|(u,v) \in \Omega_\kappa\}$ with coordinates $\Omega_k \in \Omega$ from the complete 2D DCT frequency domain $\Omega = \{(u,v)|u,v \in \{0, \ldots, d-1\}\}$.

For a parameter matrix $\Delta \boldsymbol{W} \in \mathbb{R}^{d \times d}$, the $(m, n)$-th value of $\Delta \boldsymbol{W}$ is reconstructed as

$$\Delta \boldsymbol{W}_\kappa(m, n) = \sum_{F(u,v) \in \mathcal{F}_\kappa} \alpha_u \alpha_v F(u,v) \cdot \mathbf{iDCT}_{u,v}. \tag{6}$$

where $\mathbf{iDCT}_{u,v} = \alpha_u \alpha_v \cos\left(\frac{\pi(2m+1)u}{2d}\right) \cos\left(\frac{\pi(2n+1)v}{2d}\right)$ represents the 2D iDCT basis function. $\alpha_u = \sqrt{\frac{1}{d}}$ if $u = 0$, otherwise $\alpha_u = \sqrt{\frac{2}{d}}$, and similarly for $\alpha_v$.

The critical innovation of our frequency assignment strategy is that instead of clients competing for overlapping regions in the parameter space, each client is assigned a unique subset of frequency $\mathcal{F}_\kappa$. This ensures that local model updates from different clients reside in non-overlapping frequency subspaces, making them inherently interference-free:

$$\langle \Delta \boldsymbol{W}_\kappa, \Delta \boldsymbol{W}_{\kappa'} \rangle = 0 \quad (\kappa \neq \kappa'). \tag{7}$$

For frequency aggregation, we collect all DCT coefficients: $\mathcal{F} = \{\bar{F}(u,v)|(u,v) \in \bigcup_{\kappa=1}^{K} \Omega_\kappa\}$. When multiple clients select the same frequency $(u,v)$, we average their coefficients:

$$\bar{F}(u,v) = \frac{1}{|\{\kappa : (u,v) \in \Omega_\kappa\}|} \sum_{\kappa:(u,v)\in\Omega_\kappa} F_\kappa(u,v). \tag{8}$$

The final aggregated update is reconstructed via inverse DCT(iDCT):

$$\Delta\bar{\boldsymbol{W}}(m,n) = \sum_{\bar{F}(u,v)\in\mathcal{F}} \bar{F}(u,v) \cdot \mathbf{iDCT}_{u,v}, \tag{9}$$

## 4.1 FREQUENCY ASSIGNMENT AND CLIENT INDEPENDENCE

Each client $\kappa$ randomly samples a subset of frequency coordinates $\Omega_\kappa$ from the complete frequency domain $\Omega = \{(u,v)|u,v \in \{0,\ldots,d-1\}\}$ and learns only the corresponding DCT coefficients $\mathcal{F}_\kappa = \{F_{(u,v)}|(u,v) \in \Omega_\kappa\}$. We support two assignment strategies

- **Centralized frequency assignment (strict orthogonality)**: Frequency components are allocated to clients through explicit coordination, either by a central server or through inter-client communication, to ensure strict orthogonality (no overlap between frequency subsets). Clients must communicate with the server to receive their assigned frequency coordinates before beginning local training The server maintains a global registry of which frequencies are assigned to which clients. When clients sample frequency coordinates without replacement such that $\Omega_\kappa \cap \Omega_{\kappa'} = \emptyset$ for all $\kappa \neq \kappa'$, the DCT's orthogonal basis functions guarantee:
$$\langle \Delta\boldsymbol{W}_\kappa, \Delta\boldsymbol{W}_{\kappa'}\rangle = 0, \quad \forall\kappa \neq \kappa'.$$

- **Decentralized frequency assignment (statistical orthogonality)**: Each client $\kappa$ autonomously samples a subset of frequency coordinates $\Omega_k$ from the complete frequency domain $\Omega$ using sampling with replacement. No communication required between clients or with the server beyond the final model update. and clients operate with zero knowledge of other clients' frequency selections while still achieving effective interference reduction through statistical orthogonality:
$$\mathbb{E}[\langle \Delta\boldsymbol{W}_\kappa, \Delta\boldsymbol{W}_{\kappa'}\rangle] = 0, \ \mathrm{Var}[\langle \Delta\boldsymbol{W}_\kappa, \Delta\boldsymbol{W}_{\kappa'}\rangle] = \frac{|\Omega|^2}{d^2} \cdot \sigma^4 \ (Theorem\ 4.2).$$

Both frequency assignment are feasible for `FedDCT`. however, centralized frequency assignment fundamentally violates the core principles of one-shot federated learning by requiring explicit coordination between the server and clients. It approach creates communication overhead beyond the single-round constraint, as clients must either receive frequency assignments from the server or negotiate frequency allocation among themselves, contradicting the fundamental goal of minimizing communication in one-shot federated learning. Moreover, centralized assignment suffers from poor scalability and robustness—it cannot handle dynamic client participation where clients may join or leave unexpectedly, requires the server to maintain global state about all participating clients, and introduces a single point of failure where coordination failures can compromise the entire federated learning process. The comparison of these assignment are detail in Sec. 5.2.

## 4.2 THEORETICAL ANALYSIS

We now provide theoretical guarantees of decentralized frequency assignment for `FedDCT` in one-shot federated learning.

**Theorem 4.1** (Collision Probability under Sampling with Replacement). *For clients $\kappa$ and $\kappa'$, let $\Omega_\kappa$ and $\Omega_{\kappa'}$ be independently sampled frequency sets with replacement, each containing $|\Omega_\kappa|$ and $|\Omega_{\kappa'}|$ components from a total of $d^2$ frequency components. The expected collision proportion is:*

$$\mathbb{E}\left[\frac{|\Omega_\kappa \cap \Omega_{\kappa'}|}{|\Omega_\kappa|}\right] = \frac{|\Omega_{\kappa'}|}{d^2}. \tag{10}$$

This theorem shows that even with replacement, the collision probability remains small for typical parameter settings. For example, with $d = 1024$ and $|\Omega_{\kappa'}| = 1000$, the expected collision proportion is only $\frac{1000}{1024^2} \approx 0.095\%$, meaning less than 0.1% of frequency components collide on average.

---

**Algorithm 1:** Federated One-Shot Learning via Frequency Separated Aggregation

---

1 **Require:** Each client $\mathcal{D}$ datasets $\mathcal{D}_{\kappa \in K}$, pretrained model weights $\boldsymbol{W}$, number of frequency components $N$.

2 **Ensure:** Global model $\boldsymbol{W}$

  1: **for all** client $\kappa = 1$ to $K$ **in parallel do**

  2:     Random sample $N$ frequency coordinate: $\Omega_{\kappa} = \{(\mu_n, v_n)\}_{n=1}^N$       ▷ Spectrum assignment

  3:     Initialize trainable DCT coefficient: $\mathcal{F}_{\kappa} = \{F(u,v) | (u,v) \in \Omega_{\kappa}\}$

  4:     **for** each batch training **do**

  5:         Compute $\Delta \boldsymbol{W}_{\kappa} \leftarrow \texttt{iDCT}(\mathcal{F}_{\kappa})$                      ▷ DCT inverse

  6:         Compute loss $\mathcal{L}(\boldsymbol{W} + \Delta \boldsymbol{W}_{\kappa})$ and Update $\mathcal{F}_{\kappa}$

  7:     **end for**

  8: **end for**

  9: Mix the Frequency: $\mathcal{F} = \bigcup_{\kappa=1}^K \mathcal{F}_{\kappa}$           ▷ Frequency domain aggregation

 10: Server aggregates: $\Delta \bar{\boldsymbol{W}} \leftarrow \texttt{iDCT}(\mathcal{F}_{\texttt{MixF}})$        ▷ Recover the global model

 11: **Return** updated model $\boldsymbol{W} + \Delta \bar{\boldsymbol{W}}$

---

**Theorem 4.2** (Interference Bound with Partial Collisions). *For clients $\kappa$ and $\kappa'$ with DCT coefficient sets $\mathcal{F}_{\kappa}$ and $\mathcal{F}_{\kappa'}$, let $\mathcal{C} = \Omega_{\kappa} \cap \Omega_{\kappa'}$ be the collision set. Due to the orthogonality of DCT basis functions, the interference between client updates is:*

$$\langle \Delta \boldsymbol{W}_{\kappa}, \Delta \boldsymbol{W}_{\kappa'} \rangle = \sum_{(u,v) \in \mathcal{C}} F_{\kappa}(u,v) \cdot F_{\kappa'}(u,v), \tag{11}$$

*where $F_{\kappa}(u,v)$ and $F_{\kappa'}(u,v)$ are the DCT coefficients at frequency $(u,v)$.*

*Assuming the DCT coefficients are independently distributed with zero mean and variance $\sigma^2$, the expected interference is: $\mathbb{E}[\langle \Delta \boldsymbol{W}_{\kappa}, \Delta \boldsymbol{W}_{\kappa'} \rangle] = 0$, and the variance of interference is:*

$$Var[\langle \Delta \boldsymbol{W}_{\kappa}, \Delta \boldsymbol{W}_{\kappa'} \rangle] = \mathbb{E}[|\mathcal{C}|] \cdot \sigma^4 = \frac{|\Omega_{\kappa}||\Omega_{\kappa'}|}{d^2} \cdot \sigma^4. \tag{12}$$

This theorem demonstrates that interference only occurs at colliding frequencies, and its magnitude depends on the DCT coefficients at those frequencies. The expected interference is zero due to the orthogonality of DCT basis functions, while the variance of interference scales as $\mathcal{O}(|\Omega|^2/d^2)$, which is significantly better than $\mathcal{O}(|\Omega|^2/d)$ in traditional parameter-efficient methods that operate directly in the parameter space. The quadratic denominator $d^2$ (from the 2D DCT's expanded frequency domain) ensures that interference variance remains negligible even with large frequency sets, as typical collision rates are less than 0.1% for practical parameter settings.

## 5 EXPERIMENT

**Datasets and Partitioning.** To evaluate the effectiveness of $\texttt{FedDCT}$ method, we conduct extensive experiments across diverse language understanding and generation tasks. We use three widely-adopted datasets in LLM research: Dolly-15K (Conover et al., 2023) for the question-answering (QA) task, the Rosetta dataset (Rosset et al., 2024) for code generation, and GSM-8K (Cobbe et al., 2021) for mathematical reasoning. We evaluate the federated fine-tuned models on MMLU (Hendrycks et al., 2021) and MMLU-Pro (Wang et al., 2024a) for the QA task, HumanEval (Chen et al., 2021) test set for Code generation task and GSM-8K for mathematical reasoning task, respectively. For Dolly-15K, we report accuracy, BLEU, METEOR, and ROUGE-L scores with detailed analysis. For Rosetta, we report Pass@1 and Pass@10 metrics. For GSM-8K, we report accuracy. To mimic realistic heterogeneity, the clients' dataset is partitioned via a Dirichlet distribution (Hsu et al., 2019) with concentration parameter $\alpha$; smaller values of $\alpha$ produce more skewed (i.e. heterogeneous) client splits. In our primary experiments, we set $\alpha = 0.001$ to emulate strong non-IIDness, and we additionally evaluate $\alpha \in \{0.001, 0.01, 0.05\}$ to study the impact of varying heterogeneity levels.

**Models and Training.** We conduct experiments with two foundation models: LLaMA-2-7B (Touvron et al., 2023b) and Qwen2.5-7B-Instruct (Team, 2024). All experiments use a consistent setup with

Table 1: Comparison of `FedDCT` with baseline federated learning methods across LLaMA2-7B and Qwen2.5-7B models. The results demonstrate `FedDCT` superior performance while requiring significantly lower parameter uploads (0.36M) compared to other methods (2.1–5.1M).

| Model | Method | Upload Param | MMLU | MMLU-Pro | | | | HumanEvalX | | GSM-8K |
|---|---|---|---|---|---|---|---|---|---|---|
| | | | | Acc. | BLEU | METEOR | ROUGE-L | Pass@1 | Pass@10 | |
| LLaMA2-7B | Centralised FL | 0 | $38.9_{\pm 0.1}$ | $17.6_{\pm 0.1}$ | $20.9_{\pm 0.3}$ | $28.9_{\pm 0.3}$ | $28.9_{\pm 0.4}$ | $11.6_{\pm 0.1}$ | $28.5_{\pm 0.3}$ | $17.2_{\pm 0.5}$ |
| | FLoRA (30-shots) | 2.5M | $36.4_{\pm 0.1}$ | $15.7_{\pm 0.1}$ | $21.2_{\pm 0.3}$ | $29.5_{\pm 0.8}$ | $19.9_{\pm 0.4}$ | $11.2_{\pm 0.5}$ | $15.7_{\pm 0.7}$ | $15.8_{\pm 0.2}$ |
| | **One-Shot Methods** | | | | | | | | | |
| | FLoRA (1-shot) | 2.5M | $24.6_{\pm 0.1}$ | $10.2_{\pm 0.4}$ | $03.2_{\pm 0.4}$ | $05.8_{\pm 0.6}$ | $01.5_{\pm 0.3}$ | $02.7_{\pm 0.2}$ | $06.1_{\pm 0.9}$ | $14.6_{\pm 0.9}$ |
| | DENSE | 5.1M | $26.3_{\pm 0.2}$ | $08.2_{\pm 0.2}$ | $0.01_{\pm 0.01}$ | $0.02_{\pm 0.01}$ | $0.01_{\pm 0.01}$ | $04.4_{\pm 0.1}$ | $07.1_{\pm 0.6}$ | $11.3_{\pm 0.4}$ |
| | CO-Boosting | 5.1M | $32.8_{\pm 0.4}$ | $09.5_{\pm 0.5}$ | $01.0_{\pm 0.4}$ | $0.7_{\pm 0.5}$ | $01.8_{\pm 0.3}$ | $03.2_{\pm 0.9}$ | $07.3_{\pm 0.1}$ | $12.1_{\pm 0.6}$ |
| | FuseFL | 2.5M | $34.8_{\pm 0.2}$ | $13.6_{\pm 0.2}$ | $11.1_{\pm 0.3}$ | $\mathbf{11.9}_{\pm 0.2}$ | $\mathbf{19.5}_{\pm 0.4}$ | $11.5_{\pm 0.4}$ | $16.6_{\pm 0.7}$ | $15.0_{\pm 0.3}$ |
| | FedDCT | **0.36M** | $\mathbf{35.6}_{\pm 0.2}$ | $\mathbf{14.0}_{\pm 0.7}$ | $\mathbf{13.0}_{\pm 0.1}$ | $11.7_{\pm 0.2}$ | $18.2_{\pm 0.2}$ | $\mathbf{11.9}_{\pm 0.3}$ | $\mathbf{23.1}_{\pm 0.4}$ | $\mathbf{16.2}_{\pm 0.4}$ |
| Qwen2.5-7B | Centralised FL | 0 | $69.2_{\pm 0.3}$ | $44.3_{\pm 0.3}$ | $19.4_{\pm 0.9}$ | $29.5_{\pm 0.9}$ | $20.0_{\pm 0.7}$ | $79.8_{\pm 0.8}$ | $86.6_{\pm 0.2}$ | $77.8_{\pm 0.1}$ |
| | FLoRA (30-shots) | 2.5M | $67.1_{\pm 0.4}$ | $39.6_{\pm 0.2}$ | $17.7_{\pm 0.1}$ | $26.3_{\pm 0.4}$ | $13.3_{\pm 0.7}$ | $78.2_{\pm 0.3}$ | $79.9_{\pm 0.2}$ | $77.1_{\pm 0.7}$ |
| | **One-Shot Methods** | | | | | | | | | |
| | FLoRA (1-shot) | 2.1M | $65.7_{\pm 0.3}$ | $40.0_{\pm 0.3}$ | $12.1_{\pm 0.4}$ | $\mathbf{18.6}_{\pm 0.4}$ | $13.1_{\pm 0.4}$ | $72.9_{\pm 0.6}$ | $\mathbf{84.8}_{\pm 0.9}$ | $62.6_{\pm 0.2}$ |
| | DENSE | 4.2M | $32.3_{\pm 1.0}$ | $19.1_{\pm 0.3}$ | $01.7_{\pm 0.3}$ | $04.7_{\pm 0.6}$ | $06.9_{\pm 0.1}$ | $53.7_{\pm 0.6}$ | $61.3_{\pm 0.3}$ | $52.1_{\pm 0.2}$ |
| | CO-Boosting | 4.2M | $45.3_{\pm 0.8}$ | $29.5_{\pm 0.4}$ | $01.0_{\pm 0.4}$ | $01.1_{\pm 0.1}$ | $01.8_{\pm 0.2}$ | $55.1_{\pm 0.3}$ | $74.0_{\pm 0.7}$ | $50.8_{\pm 0.4}$ |
| | FuseFL | 2.1M | $67.8_{\pm 0.4}$ | $38.7_{\pm 0.6}$ | $\mathbf{16.8}_{\pm 0.1}$ | $14.4_{\pm 0.3}$ | $16.7_{\pm 0.8}$ | $78.2_{\pm 0.2}$ | $82.7_{\pm 0.9}$ | $68.4_{\pm 0.9}$ |
| | FedDCT | **0.36M** | $\mathbf{68.6}_{\pm 0.2}$ | $\mathbf{42.3}_{\pm 0.6}$ | $16.7_{\pm 0.4}$ | $13.5_{\pm 0.9}$ | $\mathbf{16.7}_{\pm 0.3}$ | $\mathbf{79.3}_{\pm 0.1}$ | $82.3_{\pm 0.2}$ | $\mathbf{75.4}_{\pm 0.3}$ |

batch size 2 and local training for 2,000 steps per client. We perform hyperparameter tuning for the learning rate in the range $\{0.001, 0.005, 0.01, 0.05\}$ and report results using the best configuration. For our `FedDCT` method, we use the Discrete Cosine Transform (DCT) basis with the number of trainable parameter $|\Omega|(|\Omega| = 6000)$ per weight matrix. We apply our method to query and value matrices in the transformer blocks, which comprise approximately 20% of all model parameters.

**Baselines.** We compare `FedDCT` against the following baseline methods: **Centralised FL**: Centralized fine-tuning on the aggregated dataset (upper-bound reference). **FLoRA(30-shots)** (Wang et al., 2024b): Standard FedAvg aggregation for federated instruction-tuning of large language models in IID setting. **FLoRA(1-shot)**: Standard one-shot federated learning (FL) for federated instruction-tuning of large language models in Non-IID setting. **DENSE** (Zhang et al., 2022): A data-free one-shot federated learning method that alternates between synthetic data generation and model distillation to train the global model. **CO-Boosting** (Dai et al., 2024): A one-shot FL scheme in which synthesized data and an ensemble of client models progressively enhance each other. **FuseFL** (Tang et al., 2024): A one-shot federated learning approach based on model fusion, combining client updates into a single global model. For all experiments, we set the default number of clients to $M = 5$ and vary from 2 to 9 in specific scalability experiments.

## 5.1 MAIN RESULTS.

Table 1[2] presents the performance of `FedDCT` compared to baselines across different datasets and models. Our method outperforms all one-shot federated learning baselines while achieving results competitive with multi-round methods using significantly less communication. `FedDCT` consistently outperforms most one-shot FL baselines across both models and all datasets, approaching centralised performance in many cases. DENSE and CO-Boosting show lower performance, particularly for complex generation tasks, likely due to the inherent limitations of synthetic data generation. Our method demonstrates particularly strong performance on the Rosetta code generation task, where maintaining semantic coherence is critical. The gap between `FedDCT` and Centralised FL is smaller for the more capable Qwen2.5-7B model, suggesting that more powerful foundation models can better leverage our approach. When tested on LLaMA-2-7B and Qwen2.5-7B, `FedDCT` achieved scores approaching centralized training while requiring only 0.36M parameter uploads per client compared to 2.1-5.1M for these baseline methods—a 5.8-14.2× reduction in communication overhead. These findings validate the theoretical claim that `FedDCT` prevents parameter space collisions through its frequency-separated approach, enabling efficient model adaptation without the iterative communication in FLoRA or the synthetic data limitations in DENSE and CO-Boosting.

---

[2]The best results for each dataset are shown in bold.

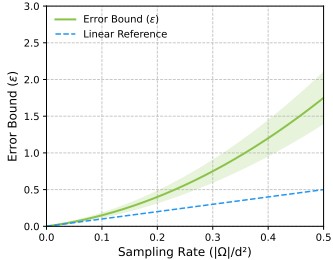 

Figure 2: Comparison of different aggregation methods for one-shot federated learning

Figure 3: Collision error bound under sampling with replacement

Figure 4: Visualization of collision patterns at different sparsity levels

## 5.2 ANALYSIS

**Analysis on Heterogeneity.** A key challenge in one-shot federated learning is handling non-IID data distributions across clients. Table 2 presents the performance of different methods under varying degrees of data heterogeneity with the Dirichlet concentration parameter $\alpha$. The experiment demonstrates FedDCT's remarkable robustness to Non-IID data distributions. When testing on the MMLU benchmark with Qwen2.5-7B, FedDCT achieves 68.6% accuracy under extreme heterogeneity ($\alpha = 0.001$), maintaining performance within 5% of centralized training (69.2%), while competing methods show significant degradation (DENSE: 32.3%, Co-Boosting: 45.3%),

Table 2: Performance under different levels of data heterogeneity (MMLU test set with Qwen2.5-7B)

| Method | $\alpha = 0.001$ | $\alpha = 0.01$ | $\alpha = 0.05$ |
|---|---|---|---|
| Centralised FL | $69.2_{\pm0.3}$ | $69.2_{\pm0.2}$ | $69.2_{\pm0.2}$ |
| DENSE | $32.3_{\pm1.0}$ | $37.6_{\pm0.9}$ | $38.8_{\pm0.5}$ |
| Co-Boosting | $45.3_{\pm0.8}$ | $45.2_{\pm0.4}$ | $54.2_{\pm0.8}$ |
| FuseFL | $67.8_{\pm0.4}$ | $67.5_{\pm0.3}$ | $67.7_{\pm0.4}$ |
| FedDCT | $\mathbf{68.6}_{\pm0.2}$ | $\mathbf{69.5}_{\pm0.4}$ | $\mathbf{68.4}_{\pm0.3}$ |

confirming that the orthogonal nature of our frequency domain transformation effectively creates separated parameter spaces, allowing clients to optimize for local data distributions without negatively impacting others during aggregation, addressing the challenge of amplified spatial heterogeneity.

**Number of Clients.** Table 3 demonstrate FedDCT's scalability as the number of clients increases in federated learning scenarios. When tested on the Dolly-15k dataset with Qwen2.5-7B, FedDCT maintains consistent performance across different client configurations, showing only a 0.9% drop in accuracy (from 69.1% to 68.2%) when scaling from 2 to 9 clients. This stability stands in stark contrast to FLoRA(1-shot), which experiences a significant 1.9% degradation (from 68.8% to 66.9%) under identical scaling conditions. Furthermore, FedDCT achieves this superior performance while requiring substantially lower communication costs - only 0.695MB for 5 clients compared to

Table 3: Performance vs. Number of Clients on Dolly-15k with Qwen2.5-7B

| Method | Clients | Hetero | Homo | Comm. Cost (MB) |
|---|---|---|---|---|
| FedDCT ($\|\Omega\| = 1000$) | 2 | $69.1_{\pm0.1}$ | $70.8_{\pm0.2}$ | 0.278 |
| | 3 | $69.2_{\pm0.4}$ | $72.4_{\pm0.5}$ | 0.417 |
| | 4 | $68.2_{\pm0.7}$ | $67.1_{\pm1.2}$ | 0.556 |
| | 5 | $68.4_{\pm0.4}$ | $69.1_{\pm0.4}$ | 0.695 |
| | 9 | $68.2_{\pm0.4}$ | $68.6_{\pm0.4}$ | 1.250 |
| FLoRA (1-shot) | 2 | $68.8_{\pm0.6}$ | $68.8_{\pm0.7}$ | 09.62 |
| | 3 | $70.2_{\pm0.2}$ | $70.1_{\pm0.3}$ | 19.24 |
| | 4 | $67.8_{\pm0.5}$ | $68.2_{\pm0.6}$ | 28.86 |
| | 5 | $65.7_{\pm0.3}$ | $68.2_{\pm0.9}$ | 48.13 |
| | 9 | $66.9_{\pm0.4}$ | $67.6_{\pm0.4}$ | 86.63 |

FLoRA's 48.13MB, representing a significant reduction in bandwidth requirements. The effectiveness of FedDCT persists in both heterogeneous and homogeneous data distribution settings, making it particularly suitable for real-world federated learning deployments.

**Centralized *vs*. Decentralized method** The results in Table 4 show that both approaches achieve similar high performance, with the decentralized method's average accuracy on the MMLU, MMLU-Pro, and GSM-8K datasets being 66.25%, 36.95%, and 71.75%, respectively, which is nearly identical to the centralized method's average scores of 66.20%, 37.85%, and 71.70%. This demonstrates that the statistical orthogonality provided by the decentralized approach is as effective as the strict orthogonality of the centralized approach. The decentralized method is more practical for real-world one-shot FL because it avoids the communication overhead, scalability issues, and single-point-of-failure risks associated with centralized coordination.

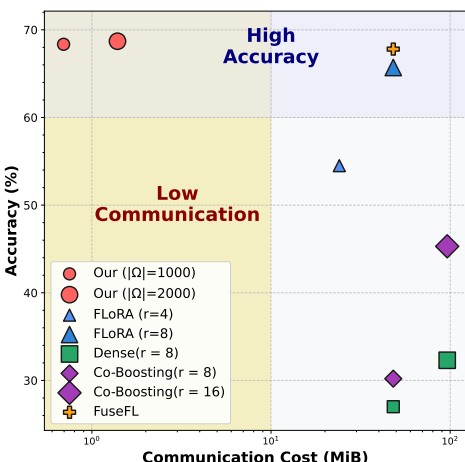

Figure 5: Trade off between communication efficiency between performance in various baselines on Llama2-7B.

| Method | Metric | 9 Clients | 40 Clients | Average |
|---|---|---|---|---|
| **Centralized** (Statistical) | MMLU | 68.2 | 64.2 | 66.20 |
| | MMLU-Pro | 41.7 | 34.0 | 37.85 |
| | GSM-8K | 74.1 | 69.3 | 71.70 |
| **Decentralized** (Strict) | MMLU | 68.2 | 64.3 | 66.25 |
| | MMLU-Pro | 41.1 | 32.8 | 36.95 |
| | GSM-8K | 74.2 | 69.3 | 71.75 |

Table 4: Performance Comparison of Centralized vs. Decentralized Frequency Assignment.

| | $|\Omega|$ | Sparsity | Acc. |
|---|---|---|---|
| FedDCT | 500 | 3.89e-5 | $66.7_{\pm 2.4}$ |
| | 1000 | 7.79e-5 | $68.4_{\pm 0.4}$ |
| | 1500 | 1.17e-4 | $68.7_{\pm 0.2}$ |
| | 2000 | 1.56e-4 | $68.6_{\pm 0.2}$ |
| | 10000 | 7.79e-4 | $68.6_{\pm 0.2}$ |
| | 12000 | 9.34e-4 | $68.8_{\pm 0.3}$ |

Table 5: Acc. v.s. $|\Omega|$. on Dolly-15k with Qwen2.5-7B.

**Type of aggregation.** As shown in Figure 2, we compare the performance of different aggregation methods for one-shot federated learning across two benchmark tasks: MMLU and GSM-8K.The results demonstrate a clear performance advantage for our proposed `FedDCT` approach over theone-shot version of FedIT(`FedAvg`) and FLoRA(`FedStack`), achieving scores of 35.8 on MMLU and 16.8 on GSM-8K. This significant improvement (approximately 57.7% higher than FedIT on MMLU and 15.0% higher than FLoRA on GSM-8K) validates our theoretical claim that frequency-separated aggregation effectively addresses parameter space collisions in one-shot federated learning.

**Collision Analysis.** Note that sparsity is controlled via the number of frequency components that each client samples from the total frequency domain. As clients sample more frequency components, sparsity increases, allowing for richer representation but potentially increasing the risk of interference between clients. Table 5 presents the relationship between the number of frequency components and model accuracy on the Dolly-15k dataset using the Qwen2.5-7B model.With just 500 frequency components (sparsity 3.90e-5), the model achieves a respectable 66.7% accuracy. Doubling this to 1000 components (sparsity 7.70e-5) yields a significant improvement to 68.7% accuracy, representing the highest performance observed across all configurations. Further increases in frequency components produce minimal benefits or even slight decreases in performance: 1500 components (68.7%), 2000 components (68.6%), 10000 components (68.6%), and 12000 components (68.8%). The table show that there exists an optimal range for frequency component allocation. Too few components limit expressivity, while too many introduce redundancy and potential overfitting. Figure 4 further visualizes collision patterns at different sparsity levels.

**Communication Efficiency Analysis.** Figure 5 illustrates the trade-off between communication cost and model performance for different federated learning methods using the LLaMA2-7B model. Results shows that `FedDCT` variants ($|\Omega|$=1000 and $|\Omega|$=2000) occupy the optimal upper-left region, indicating high accuracy with minimal communication requirements. This empirical result directly validates our theoretical analysis that frequency-separated aggregation effectively creates almost-orthogonal parameter spaces, enabling efficient knowledge transfer without the massive communication burden of other one-shot federated learning approaches.

## 6 CONCLUSION

We identified parameter space interference as a major challenge in one-shot federated learning. We theoretically proved that using orthogonal parameter spaces effectively eliminates this interference. Our proposed `FedDCT` approach leverages orthogonal transformations in the frequency domain to achieve this naturally, allowing clients to learn in non-overlapping regions without explicit coordination. By sampling frequency components with replacement, we create an almost-orthogonal parameter space, which enables efficient knowledge transfer and perfect global model reconstruction while maintaining parameter efficiency.

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

# A    APPENDIX

## A.1    THE USE OF LARGE LANGUAGE MODELS

We use LLMs for only polish the writing. No LLMs are used for creating idea, code and other artifacts.

## A.2    PROOF OF THEOREM 4.1

*Proof.* Consider two clients $\kappa$ and $\kappa'$ independently sampling frequency components with replacement from the total frequency domain of size $d^2$.

For any specific frequency coordinate $(u, v)$, the probability that client $\kappa$ selects it is:

$$P((u, v) \in \Omega_\kappa) = \frac{|\Omega_\kappa|}{d^2} \tag{13}$$

Similarly, for client $\kappa'$:

$$P((u, v) \in \Omega_{\kappa'}) = \frac{|\Omega_{\kappa'}|}{d^2} \tag{14}$$

Since the clients sample independently, the probability that both select the same frequency $(u, v)$ is:

$$P((u, v) \in \Omega_\kappa \cap \Omega_{\kappa'}) = P((u, v) \in \Omega_\kappa) \cdot P((u, v) \in \Omega_{\kappa'}) = \frac{|\Omega_\kappa||\Omega_{\kappa'}|}{d^4} \tag{15}$$

The expected number of collisions is the sum over all possible frequency coordinates:

$$\mathbb{E}[|\Omega_\kappa \cap \Omega_{\kappa'}|] = \sum_{u=0}^{d-1} \sum_{v=0}^{d-1} P((u, v) \in \Omega_\kappa \cap \Omega_{\kappa'}) = d^2 \cdot \frac{|\Omega_\kappa||\Omega_{\kappa'}|}{d^4} = \frac{|\Omega_\kappa||\Omega_{\kappa'}|}{d^2} \tag{16}$$

Therefore, the expected collision proportion is:

$$\mathbb{E}\left[\frac{|\Omega_\kappa \cap \Omega_{\kappa'}|}{|\Omega_\kappa|}\right] = \frac{\mathbb{E}[|\Omega_\kappa \cap \Omega_{\kappa'}|]}{|\Omega_\kappa|} = \frac{|\Omega_\kappa||\Omega_{\kappa'}|/d^2}{|\Omega_\kappa|} = \frac{|\Omega_{\kappa'}|}{d^2} \tag{17}$$

$\square$

## A.3    PROOF OF THEOREM 4.2

*Proof.* Given the DCT representation of client updates:

$$\Delta \boldsymbol{W}_\kappa = \sum_{(u,v) \in \Omega_\kappa} F_\kappa(u, v) \cdot \mathbf{iDCT}_{u,v} \tag{18}$$

The inner product between two client updates is:

$$\langle \Delta \boldsymbol{W}_\kappa, \Delta \boldsymbol{W}_{\kappa'} \rangle = \left\langle \sum_{(u,v) \in \Omega_\kappa} F_\kappa(u, v) \cdot \mathbf{iDCT}_{u,v}, \sum_{(u',v') \in \Omega_{\kappa'}} F_{\kappa'}(u', v') \cdot \mathbf{iDCT}_{u',v'} \right\rangle$$
$$= \sum_{(u,v) \in \Omega_\kappa} \sum_{(u',v') \in \Omega_{\kappa'}} F_\kappa(u, v) \cdot F_{\kappa'}(u', v') \cdot \langle \mathbf{iDCT}_{u,v}, \mathbf{iDCT}_{u',v'} \rangle \tag{19}$$

Due to the orthogonality of DCT basis functions:

$$\langle \mathbf{iDCT}_{u,v}, \mathbf{iDCT}_{u',v'} \rangle = \begin{cases} 1 & \text{if } (u, v) = (u', v') \\ 0 & \text{otherwise} \end{cases} \tag{20}$$

Therefore:

$$\langle \Delta \boldsymbol{W}_\kappa, \Delta \boldsymbol{W}_{\kappa'} \rangle = \sum_{(u,v) \in \Omega_\kappa \cap \Omega_{\kappa'}} F_\kappa(u, v) \cdot F_{\kappa'}(u, v) \tag{21}$$

For the expected value, assuming $F_\kappa(u, v)$ and $F_{\kappa'}(u, v)$ are independent with zero mean:

$$\mathbb{E}[F_\kappa(u, v) \cdot F_{\kappa'}(u, v)] = \mathbb{E}[F_\kappa(u, v)] \cdot \mathbb{E}[F_{\kappa'}(u, v)] = 0 \cdot 0 = 0 \tag{22}$$

Thus:

$$\mathbb{E}[\langle \Delta \boldsymbol{W}_\kappa, \Delta \boldsymbol{W}_{\kappa'} \rangle] = \sum_{(u,v) \in \Omega_\kappa \cap \Omega_{\kappa'}} \mathbb{E}[F_\kappa(u, v) \cdot F_{\kappa'}(u, v)] = 0 \tag{23}$$

For the variance, since the coefficients are independent across frequencies:

$$\mathrm{Var}[\langle \Delta \boldsymbol{W}_\kappa, \Delta \boldsymbol{W}_{\kappa'} \rangle] = \mathrm{Var}\left[ \sum_{(u,v) \in \mathcal{C}} F_\kappa(u, v) \cdot F_{\kappa'}(u, v) \right]$$
$$= \sum_{(u,v) \in \mathcal{C}} \mathrm{Var}[F_\kappa(u, v) \cdot F_{\kappa'}(u, v)] \tag{24}$$

For independent random variables with variance $\sigma^2$:

$$\mathrm{Var}[F_\kappa(u, v) \cdot F_{\kappa'}(u, v)] = \mathbb{E}[(F_\kappa(u, v) \cdot F_{\kappa'}(u, v))^2] = \mathbb{E}[F_\kappa(u, v)^2] \cdot \mathbb{E}[F_{\kappa'}(u, v)^2] = \sigma^2 \cdot \sigma^2 = \sigma^4 \tag{25}$$

Therefore:

$$\mathrm{Var}[\langle \Delta \boldsymbol{W}_\kappa, \Delta \boldsymbol{W}_{\kappa'} \rangle] = \mathbb{E}[|\mathcal{C}|] \cdot \sigma^4 = \frac{|\Omega_\kappa||\Omega_{\kappa'}|}{d^2} \cdot \sigma^4 \tag{26}$$

where we used the result from Theorem 4.1 that $\mathbb{E}[|\mathcal{C}|] = \frac{|\Omega_\kappa||\Omega_{\kappa'}|}{d^2}$. $\qquad\square$

# B EXPERIMENT

## B.1 DETAILS

We trained our model with two NVIDIA A100 GPUs (each with 39 GB of memory). Training took approximately 1.5 hours, while testing required about 2 hours. We reproduced all baselines because they are adopted other datasets and models.

Metic: **BLEU (Bilingual Evaluation Understudy)**: This metric compares the machine-generated text to one or several reference texts by checking how many n-grams (word sequences) they share. **METEOR (Metric for Evaluation of Translation with Explicit ORdering)**: Matches words from the generated output to the reference text and calculates a score that combines precision and recall, taking synonyms and word stems into account. **ROUGE-L (Recall-Oriented Understudy for Gisting Evaluation)**: Measures the longest common subsequence between the generated and reference summaries, which is useful for evaluating paraphrased or rephrased text.

## B.2 DETAILS OF DATASETS

**Dolly-15K dataset**(Conover et al., 2023) An open-source collection of 15,000 text samples generated by Databricks employees. It covers brainstorming, classification, closed-QA, generation, information extraction, open-QA, and summarization. Data are partitioned by category, and we randomly sample 1,000 for a concise yet comprehensive evaluation.
For example:

- **Instruction**: When did Virgin Australia start operating?

- **Context**: ......

- **Response**:......

- **Category**:closed_qa

**GSM8K (Grade School Math 8K)**(Cobbe et al., 2021) A set of 8,500 high-quality, linguistically diverse grade-school math word problems designed for multi-step reasoning. Partitioned by problem topic.
For example:

- **Question:**......

- **Answer:**......

- **Category**:Data interpretation

**Rosetta Code**(Rosset et al., 2024) A programming chrestomathy site presenting solutions to the same task in many languages to highlight similarities and differences. Partitioned by programming language.

- **Task description:**......

- **Code:**......

- **Category**:Data interpretation

**MMLU and MMLU-Pro test set**(Hendrycks et al., 2021) MMLU contains 14,024 multiple-choice questions across 57 subjects to evaluate LLM reasoning and MMLU-Pro. We randomly sample 1,000 for a concise yet comprehensive evaluation. We adopt a one-shot setting(Just given one example) on MMLU and a five-shots on MMLU-Pro.

**HumanEval test set**(Chen et al., 2021) Released by OpenAI, comprises 164 programming problems with function signatures, docstrings, templates, and unit tests. Handcrafted to avoid overlap with

training data.

## C  LIMITATION

Our experiments focus on NLP tasks (QA, code, math reasoning) and two 7 B-parameter models. The effectiveness of `FedDCT` on larger models (e.g. 100B+), other modalities (e.g. vision or multimodal transformers), or different fine-tuning objectives remains to be validated. Because of limited computation resources, we cannot test on the larger model.

