# OpenReview forum: "Towards Interference-Free One-Shot Federated Learning via Frequency Domain Aggregation"
_ICLR.cc/2026/Conference — ICLR 2026 Conference Withdrawn Submission_

### Official Review · Reviewer_khpG · 2025-10-30

**Soundness:** 2
**Presentation:** 2
**Contribution:** 2
**Rating:** 4
**Confidence:** 5

**Summary:**

The paper _“FedDCT: Towards Interference-Free One-Shot Federated Learning via Frequency Domain Aggregation”_ proposes a frequency-domain approach for federated model aggregation.  By applying the Discrete Cosine Transform (DCT) to local model parameters, each client updates a distinct subset of frequency components, which are orthogonal in the algebraic sense.  The goal is to mitigate _parameter interference_ in one-shot federated learning by ensuring that different clients operate in mutually independent subspaces. The method leverages orthogonality and energy compaction properties of DCT, and experimental results demonstrate performance improvements compared to existing one-shot FL baselines.

**Strengths:**

**S1. Novel frequency-domain perspective on FL aggregation**
- Introducing DCT-based parameter transformation in the federated context is an original idea.
- It provides a new lens to view communication-efficient or one-shot FL through the frequency decomposition of model parameters.

**S2. Mathematical clarity and formal orthogonality proof**
- The paper clearly demonstrates that DCT yields orthogonal parameter bases, and the derivations (e.g., Eq. 5–7) are mathematically rigorous at the algebraic level.
- This formal structure gives the method a solid linear-algebraic foundation.

**S3. Practical simplicity and compatibility**
- The approach requires no architectural modification to clients or server models.
- It can be easily integrated with standard FL protocols and supports arbitrary model architectures.

**Weaknesses:**

**W1. On the Definition and Essence of “Parameter Interference”**

The paper identifies “parameter interference” as the core challenge of one-shot FL, yet the concept remains **qualitative and intuitively motivated**.  While the text mentions “amplified spatial heterogeneity” and “parameter collisions,” noquantitative metric is introduced to measure the severity or practical impact of such interference. Moreover, **parameter-space overlap** does not necessarily imply **gradient-space conflict**: even overlapping parameter updates may still contribute constructively to loss reduction if their gradients are aligned.   Therefore, it is unclear whether _parameter interference_ truly reflects detrimental gradient interactions or simply coincidental weight overlap.

**W2. From Algebraic Orthogonality to Gradient Independence**

The proposed FedDCT method transforms model parameters into the frequency domain and lets each client update disjoint frequency components.  This operation ensures _algebraic orthogonality_ among clients’ updates—essentially, each client modifies a non-overlapping subset of parameters.  However, since the model output is determined by the joint interaction of all parameters, such algebraic independence does not necessarily imply _semantic_ or _functional_ independence.  Even if different clients update different frequency components, their updates may still interfere in the functional or gradient space due to nonlinear parameter coupling.

**Questions:**

**Q1. Could the authors clarify whether parameter overlap is indeed the fundamental cause of update conflict?  Is it possible that two clients exhibit parameter overlap yet remain gradient-aligned (thus non-conflicting)?**
A clearer definition—perhaps via measurable quantities such as _update overlap ratio_ or _gradient cosine similarity_—would make the notion of “parameter interference” more rigorous and testable.

**Q2. Could the authors provide a theoretical justification or empirical evidence showing how DCT decomposition actually reduces gradient interference or directional conflicts in the optimization space, beyond the algebraic orthogonality of parameter updates?**

---

### Official Review · Reviewer_rJQN · 2025-10-30

**Soundness:** 3
**Presentation:** 3
**Contribution:** 2
**Rating:** 4
**Confidence:** 5

**Summary:**

This paper tackles the high communication cost of applying federated learning (FL) to large language models (LLMs). The authors propose FedDCT, a one-shot federated learning (OFL) framework that leverages the Discrete Cosine Transform (DCT) to decompose model parameters into orthogonal frequency subspaces. Each client operates on a distinct or statistically orthogonal subset of frequency components, effectively mitigating parameter interference during aggregation.

**Strengths:**

1. Using the DCT to construct orthogonal parameter subspaces is an original and technically meaningful idea. By expanding the parameter space quadratically (from d×d to d²), the method theoretically suppresses client collisions.
2. The paper evaluates FedDCT on diverse LLM tasks (QA, code generation, reasoning) with multiple metrics and systematically studies heterogeneity and scalability. Results consistently confirm the method’s robustness to non-IID data.
3. The decentralized frequency assignment mechanism allows clients to operate fully independently without coordination, satisfying the strict communication limits of one-shot FL.

**Weaknesses:**

1. The authors should provide a clearer theoretical justification for why operating in the frequency domain can effectively eliminate parameter interference. While the paper attributes this benefit to the orthogonality of the DCT basis, it does not explain why DCT specifically is superior in this regard. A comparative analysis or theoretical reasoning contrasting DCT with other orthogonal transforms such as Fourier or randomized orthogonal mappings would strengthen the claim.

2. The authors propose using DCT to address parameter interference. However, to my knowledge, similar techniques have already been explored in federated learning. How does the proposed method differ from or improve upon these existing approaches?
[1] FedFT: Improving Communication Performance for Federated Learning with Frequency Space Transformation
[2] DRIFT: DCT-based Robust and Intelligent Federated Learning with Trusted Privacy

3. Since the study emphasizes communication efficiency, it would be informative to compare FedDCT with few-round or quantized FL schemes to contextualize its advantage.

4. Experiments are conducted on 7B-scale models with short local training (2,000 steps). It remains unclear whether the proposed approach maintains numerical stability and computational efficiency on 30B+ models.

5. Theorem 4.2 assumes that F_k (u,v) are independent with zero mean. However, in practice, LoRA update matrices often exhibit highly correlated low-frequency components. I wonder whether the assumption Eq.23 would still hold if these frequency components are concentrated in similar regions.

6. The introduction lacks a motivation for the use of the 2D Discrete Cosine Transform (DCT). Additionally, the related work section does not include a discussion or overview of DCT-based methods.

**Questions:**

See Weaknesses

---

### Official Review · Reviewer_jg1E · 2025-11-07

**Soundness:** 3
**Presentation:** 3
**Contribution:** 2
**Rating:** 4
**Confidence:** 4

**Summary:**

This paper addresses the performance degradation problem in one-shot federated learning for large language models, which arises from parameter interference during the aggregation of client updates. To tackle this, the authors propose FedDCT, a frequency separation aggregation method based on the Discrete Cosine Transform (DCT). FedDCT assigns orthogonal frequency components to each client, ensuring that the parameter update will not interfere with one another in the frequency domain. Experiments on multiple benchmarks show that FedDCT outperforms existing methods, validating its effectiveness.

**Strengths:**

(1) The motivation of the paper is sound, which leverages frequency domain orthogonality to prevent interference among client weights during aggregation.
(2) The method effectively reduces the communication overhead of federated learning while maintaining strong model performance, addressing a critical bottleneck in the field.
(3) The paper is well-structured and clearly written, making the proposed method and experimental results easy to follow.

**Weaknesses:**

(1) The paper lacks a reason for choosing DCT. An ablation study comparing DCT with alternatives, such as Discrete Fourier Transform (DFT) or Discrete Wavelet Transform (DWT), may benefit from demonstrating the superiority of DCT.
(2) The paper is missing a theoretical analysis of communication complexity.
(3) The authors should provide a analysis of the computational cost overhead introduced by the DCT/IDCT operations on both the client and server sides.
(4) This paper is limited to the scope of NLP tasks. Authors should extend their work to other domains to prove their applicability, including CV tasks, for comparison with prior work such as DENSE[1].
(5) A comparison with methods like FLoRA[2] over 30-shot updates would show the method's capabilities and limitations better.
Ref:
[1] DENSE: Data-Free One-Shot Federated Learning, NeurIPS 2022.
[2] FLoRA: Federated Fine-Tuning Large Language Models with Heterogeneous Low-Rank Adaptations, NeurIPS 2024.

**Questions:**

see weaknesses

---

### Note · Authors · 2026-01-18

I have read and agree with the venue's withdrawal policy on behalf of myself and my co-authors.